# MoBE: Mixture-of-Basis-Experts for Compressing MoE-based LLMs

**Xiaodong Chen**[2,1*], **Mingming Ha**[1], **Zhenzhong Lan**[1,3], **Jing Zhang**[2†], **Jianguo Li**[1†]

[1]Inclusion AI, China
[2]School of Information, Renmin University of China, Beijing, China
[3]Westlake University, Hangzhou, China

{chenxiaodong, zhang-jing}@ruc.edu.cn, lijg.zero@antgroup.com

## ABSTRACT

The Mixture-of-Experts (MoE) architecture has become a predominant paradigm for scaling large language models (LLMs). Despite offering strong performance and computational efficiency, large MoE-based LLMs like DeepSeek-V3-0324 and Kimi-K2-Instruct present serious challenges due to substantial memory requirements in deployment. While recent works have explored MoE compression to address this issue, existing methods often suffer from considerable accuracy drops (e.g., 7-14% relatively) even at modest compression rates. This paper introduces a novel Mixture-of-Basis-Experts (MoBE) method that achieves model compression while incurring minimal accuracy drops. Specifically, each up/gate matrix in an expert is decomposed via a rank decomposition as $\mathbf{W} = \mathbf{AB}$, where matrix $\mathbf{A}$ is unique to each expert. The relatively larger matrix $\mathbf{B}$ is further reparameterized as a linear combination of basis matrices $\{B^i\}$ shared across all experts within a given MoE layer. The factorization is learned by minimizing the reconstruction error relative to the original weight matrices. Experiments demonstrate that MoBE achieves notably lower accuracy drops compared to prior works. For instance, MoBE can reduce the parameter counts of Qwen3-235B-A22B-2507, DeepSeek-V3-0324 (671B) and Kimi-K2-Instruct (1T) by 24%-30% with only 1%-2% accuracy drop (about 2% drops when measured relatively).

## 1 INTRODUCTION

Transformer-based large language models (LLMs) (Vaswani et al., 2017) have revolutionized natural language processing, achieving state-of-the-art performance in domains such as creative writing, code generation, and mathematical reasoning. This progress has been largely guided by scaling laws (Kaplan et al., 2020; Hoffmann et al., 2022), which posit that model performance improves with increases in parameter count and training data size. However, scaling dense architectures beyond a certain threshold—typically hundreds of billions of parameters (>100B)—has proven challenging and prohibitive. Therefore, the Mixture-of-Experts (MoE) (Jacobs et al., 1991; Jordan & Jacobs, 1994; Cai et al., 2024) architecture has become popular since the sparse activation makes MoEs much easier and more efficient to scale to more than several hundreds of billions of parameters (Liu et al., 2024; Yang et al., 2025; Team et al., 2025a) since last year.

Despite the computational advantages of sparse activation, the large total parameter counts of MoE-based LLMs present a significant bottleneck for practical deployment. For instance, leading open-source LLMs such as DeepSeek-V3-0324 (671B parameters) (Liu et al., 2024) exhibit performance comparable to top closed-source models. However, their scale imposes prohibitive demands on GPU memory; even high-end infrastructure, such as a machine with 8x H100 GPUs, may be insufficient for efficient inference.

---

* This work was done when Xiaodong Chen was a research intern at Ant Group.
† Corresponding authors.

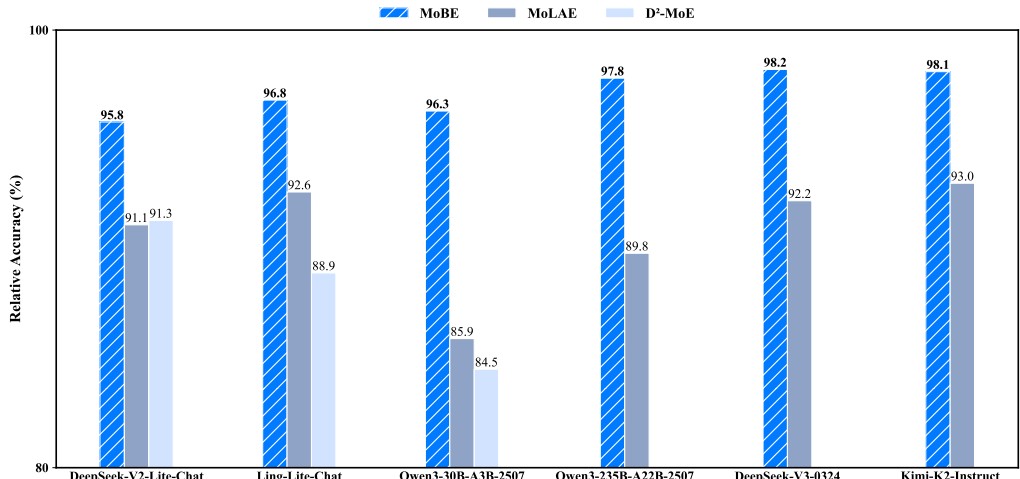

Figure 1: Relative performance comparison of different MoE compression methods. Relative accuracy is the ratio of the compressed model's performance to that of the original model. The accuracy are averaged over 15 benchmarks as shown in Table 3. Applying $D^2$-MoE to large models like Qwen3-235B-A22B-2507, DeepSeek-V3-0324 and Kimi-K2-Instruct is computationally prohibitive on an 8x H100 GPU machine; therefore, it is excluded from these comparisons. MoBE is evaluated at compression rates similar to or higher than the baseline methods (MoLAE, $D^2$-MoE).

To address this challenge, much research have been proposed for MoE-based LLM compression, which could be generally categorized into two major categories. *First*, pruning techniques reduce total parameter counts by either removing entire experts (Xie et al., 2024; Lu et al., 2024; Yang et al., 2024) or merging similar ones (hao Liu et al., 2024; Li et al., 2023b; Chen et al., 2024). However, this approach often leads to a permanent loss of specialized knowledge and significant performance degradation (Gu et al., 2025). *Second*, decomposition techniques employ matrix factorization to compress each expert's weight matrices (Gu et al., 2025; Liu et al., 2025; Li et al., 2025b). Typical works include $D^2$-MoE (Gu et al., 2025), which extracts shared weights and applies singular value decomposition (SVD) to the residual delta weights, and MoLAE (Liu et al., 2025), which uses SVD to represent each expert weight as a product of its unique transformation matrix and a shared latent matrix. Although these SVD-based methods generally outperform expert pruning, they can still incur substantial information loss. This is evidenced by the high Mean Squared Error (MSE) between the original and reconstructed matrices, as shown in our reconstruction error analysis (Figure 2).

In this paper, we introduce the Mixture-of-Basis-Experts (MoBE), a novel method for efficient, performance-preserving parameter compression for MoE-based LLMs. MoBE factorizes weight matrix $\mathbf{W}$ in an expert with rank decomposition $\mathbf{W} = \mathbf{AB}$, where $\mathbf{A}$ is unique for each expert and $\mathbf{B}$ is re-parameterized as a linear combination of a set of basis matrices $\{B^i\}$ that are shared across all experts within each MoE layer. This formulation achieves parameter reduction for two reasons. First, the number of basis matrices $m$ is much smaller than the number of experts $n$, i.e. $m \ll n$, and basis $\{B^i\}$ is shared across all experts within each layer so that we could save considerable parameters for $\mathbf{B}$. Second, the unique transformation matrix $\mathbf{A}$ is smaller than $\mathbf{W}$, so that the whole MoBE factorization achieves parameter savings. The MoBE factorization is optimized by minimizing the reconstruction error between the factorized representation and the original pretrained weight matrices, typically using the gradient descent method.

We conduct comprehensive experiments on a diverse set of MoE-based LLMs, including Ling-Lite-Chat (Team et al., 2025b), DeepSeek-V2-Lite-Chat (Shao et al., 2024), DeepSeek-V3-0324 (Liu et al., 2024), Qwen3-30B-A3B-2507, Qwen3-235B-A22B-2507 (Yang et al., 2025) and Kimi-K2-Instruct (Team et al., 2025a). A direct comparison of reconstruction error on Qwen3-30B-A3B-2507 demonstrates that MoBE achieves a consistently lower MSE than both MoLAE and $D^2$-MoE, often with reductions of over 50%, across all layers (Figure 2). Similar results for more models are presented in Appendix C. To assess downstream task performance, we evaluate the compressed models

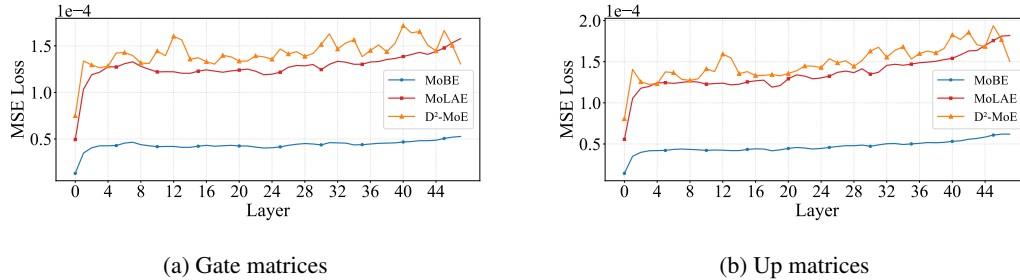

(a) Gate matrices

(b) Up matrices

Figure 2: Comparison of per-layer MSE loss for compressing the gate (a) and up (b) matrices of Qwen3-30B-A3B-2507 using MoBE, $D^2$-MoE and MoLAE.

on a wide range of benchmarks. As shown in Figure 1, MoBE exhibits a superior performance advance compared to MoLAE and $D^2$-MoE at similar or even higher compression rates.

In summary, our contributions can be summarized as follows:

- We introduce the Mixture-of-Basis-Experts (MoBE), a parameter-efficient architecture for MoE model compression. Our analysis shows that this design yields significantly lower reconstruction error compared to existing decomposition techniques.

- We demonstrate through extensive experiments on leading MoE models, including Qwen3-235B-A22B-2507, DeepSeek-V3-0324 and Kimi-K2-Instruct, that MoBE can reduce total parameter counts by 24%-30% while retaining up to 98% of the original performance, outperforming state-of-the-art MoE counterparts by a large margin.

## 2 RELATED WORKS

Research on MoE compression can be categorized into expert pruning-based (Xie et al., 2024; Lu et al., 2024; Yang et al., 2024) and decomposition-based (Li et al., 2025b; Liu et al., 2025; Gu et al., 2025). Below we elaborate on related works under these two categories.

### 2.1 EXPERT PRUNING-BASED MoE COMPRESSION METHODS

Expert pruning-based methods aim to reduce the total parameter counts of MoE-based LLMs by either directly removing entire experts or merging them. For instance, NAEE (Lu et al., 2024) removes unimportant experts by evaluating expert combinations on a calibration dataset to minimize model loss, while STUN (Lee et al., 2024) groups experts based on co-activation frequency and routing weight similarity, retaining only one expert per group. Other approaches focus on merging similar experts. DEK (Zhang et al., 2024), for example, identifies and groups similar experts in the feature space and then merges them in the weight space to reduce redundancy. MC-SMoE (Li et al., 2023b) organizes experts into distinct groups according to routing strategies and merges each group into a single expert. Because these methods remove entire expert modules, they risk a permanent loss of specialized knowledge, often leading to notable accuracy degradation on certain tasks.

### 2.2 EXPERT MATRIX DECOMPOSITION-BASED MoE COMPRESSION METHODS

In contrast to expert pruning, expert matrix decomposition-based methods compress MoE-based LLMs by factorizing each expert's weight matrices into relatively smaller representations. $D^2$-MoE (Gu et al., 2025) and MoLAE (Liu et al., 2025) are two state-of-the-art examples of this category. $D^2$-MoE approximates each expert matrix with a shared matrix and a residual delta matrix, in which the shared weight is obtained via a Fisher-weighted average of the original weights, and the residual delta weights (the difference between original and shared weights) are decomposed into low-rank matrices using SVD. MoLAE first groups a set of up/gate matrices in each MoE layer, and then approximates each matrix in a group by an expert-specific transformation matrix and the

product of a group-shared latent matrix. The approximation is achieved using SVD on the stacked up/gate matrices within the group.

Although these methods are effective in reducing parameter counts, their reliance on low-rank assumptions can be a limitation. The resulting matrix factorization does not always capture the full information of the original weights, which can introduce substantial reconstruction errors and lead to notable performance drops in downstream tasks. In Appendix B, we analyze the effective rank of expert weight matrices in several leading open-source MoE models. Our results show that this rank consistently exceeds the compression threshold of SVD—meaning that to achieve parameter reduction, the number of retained singular values must fall below this threshold. Eliminating this excess rank reduces the matrix's expressive power, likely explaining the performance degradation observed in these SVD-based compression methods.

## 3 METHODOLOGY

In this section, we first briefly review the standard Mixture-of-Experts (MoE) architecture (Section 3.1). Then, we elaborate our proposed Mixture-of-Basis-Experts (MoBE) architecture and detail the algorithm for converting a pretrained MoE model to MoBE architecture (Section 3.2). Finally, we describe the activation functions in MoBE (Section 3.3) and a specific Z-score normalization technique applied to the expert weight matrices during the conversion process (Section 3.4).

### 3.1 STANDARD MIXTURE-OF-EXPERTS ARCHITECTURE

A standard MoE layer replaces the dense Feed-Forward Network (FFN) in the Transformer with a sparsely activated structure comprising a router and multiple experts. For each input token, the router dynamically selects a small subset of these experts for processing, which yields significant computation cost reduction. In a typical MoE layer with $n$ experts, the $i$-th expert ($E^i$) often employs a SwiGLU formulation (Shazeer, 2020) to process an input token embedding $x \in \mathbb{R}^d$ as

$$E^i(x) = W^i_{down} \cdot (W^i_{up}x \odot \text{SiLU}(W^i_{gate}x)), \tag{1}$$

where $W^i_{up/gate} \in \mathbb{R}^{p \times d}$ and $W^i_{down} \in \mathbb{R}^{d \times p}$ denote the up, gate, and down projection matrices of $E^i$, $p$ is the intermediate dimension of MoE experts, and $d$ is the hidden dimension of the model. It is observed in most open-source MoE models that $p < \frac{1}{2}d$. The router $G$ calculates a gating score for each expert and selects the top-K experts for the token:

$$G(x) = \text{TopK}(\text{Softmax}(W_g x)) \tag{2}$$

where $W_g \in \mathbb{R}^{n \times d}$ denotes the weight matrix of the router $G$. The final output $y$ of the MoE layer is a weighted sum of the outputs from the selected experts:

$$y = \sum_{i=1}^{K} G^i(x)E^i(x), \tag{3}$$

where $G^i(x)$ denotes the gating value (i.e., the router score) of the $i$-th expert $E^i$. This operation is applied independently to every token in the input sequence.

### 3.2 MIXTURE-OF-BASIS-EXPERTS ARCHITECTURE

While large MoE models are much more efficient in inference than dense models of a similar size, they are also constrained by higher memory and storage requirements during deployment. To alleviate this, we introduce the Mixture-of-Basis-Experts (MoBE) architecture, as illustrated in Figure 3. The MoBE formulation begins by factorizing the up/gate matrix $W^i \in \mathbb{R}^{p \times d}$ of the $i$-th expert from the perspective of rank decomposition (Golub & Van Loan, 2013) as

$$W^i = A^i \mathbf{B}^i,$$

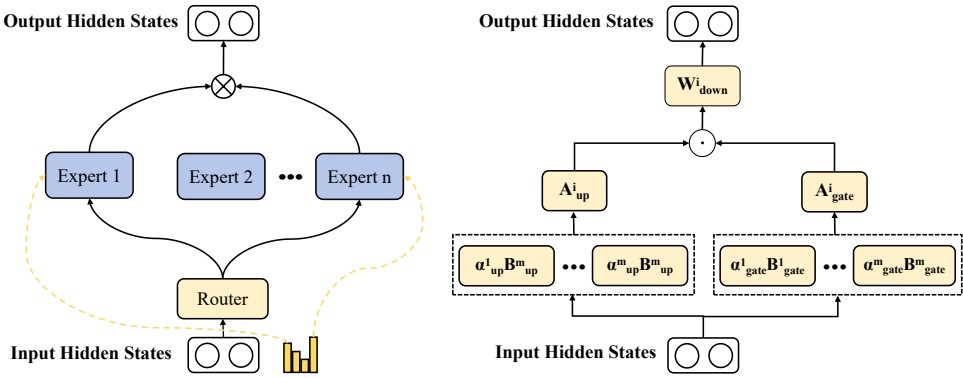

Figure 3: The Mixture-of-Basis-Experts (MoBE) architecture. For clarity of explanation, we omit the activation function following the gate matrix.

where $A^i \in \mathbb{R}^{p \times r}$, $\mathbf{B}^i \in \mathbb{R}^{r \times d}$, and $r$ is the rank of $W_i$ with $r \leq \min\{p, d\} = p$. MoBE further considers re-parameterizing $\mathbf{B}^i$ with a set of shared basis matrices as

$$\mathbf{B}^i = \sum_{j=1}^{m} \alpha^{i,j} B^j,$$

$$\text{with} \quad \alpha^{i,j} \geq 0, \quad \sum_{j=1}^{m} \alpha^{i,j} = 1,$$

where $\{B^j \in \mathbb{R}^{r \times d}\}_{j=1}^{m}$ is a set of basis matrices shared in one MoE layer, and $\{\alpha^{i,j}\}_{j=1}^{m}$ are learnable, expert-specific weighted coefficients. Combining these components and introducing a non-linear activation function $f$ (e.g., SiLU (Ramachandran et al., 2018)) to enhance representational power, we define the final MoBE factorization as:

$$\hat{W}^i = A^i f(\sum_{j=1}^{m} \alpha^{i,j} B^j), \tag{4}$$

where $\hat{W}^i$ is the reconstructed version of $W^i$.

This factorization allows the shared basis matrices $\{B^j\}$ to capture common information across all experts in one layer, while the expert-specific transformation matrices $A^i$ encode specialized information. We demonstrate in the Appendix D that this factorization is more powerful than the simple SVD approach. We apply this factorization to both the gate and up projection matrices. However, we do not decompose the down projection matrices, as prior research indicates they store critical knowledge (Geva et al., 2020; Meng et al., 2022) and are less amenable to effective compression (Liu et al., 2025).

We convert a pretrained MoE-based LLM into our proposed MoBE formulation by learning the factorized components. This is achieved by minimizing the reconstruction error between the original expert weight matrix $W^i$ and the reconstruction matrix $\hat{W}^i$ as

$$\min_{A^i, B^j, \alpha^{i,j}} \sum_{i=1}^{n} \left\| W^i - \hat{W}^i \right\|^2 = \sum_{i=1}^{n} \left\| W^i - A^i f(\sum_{j=1}^{m} \alpha^{i,j} B^j) \right\|^2 \tag{5}$$

This optimization problem can be solved using various algorithms, such as gradient-based optimizers like Adam (Kingma & Ba, 2014) or the Alternating Optimization (AO) method (Wu & Lange, 2008). In our practice, we find that the Adam optimizer performs sufficiently well across layers and various models, while AO suffers from unstable behavior during its alternating optimization steps. Algorithm 1 details the full procedure for converting a standard MoE model to the MoBE formulation.

---

**Algorithm 1** Converting standard MoE into MoBE

---

1: **Require:** $L$-layers model $\mathcal{M}_{\text{MoE}}$ with $n$ experts per layer; target basis count $m \ll n$; activation function $f$.
2: **Ensure:** Parameter-efficient MoBE model $\mathcal{M}_{\text{MoBE}}$.

3: Initialize non-MoE parts in $\mathcal{M}_{\text{MoBE}}$ with parameters directly from $\mathcal{M}_{\text{MoE}}$.
4: **for** each MoE layer $l \leq L$ in $\mathcal{M}_{\text{MoE}}$ **do**
5:     **for** type $t \in \{\text{gate, up}\}$ **do**
6:         Let $\{W_t^i\}_{i=1}^n$ be the expert matrices of the $l$-th layer
7:         Solve Eq(5) with Adam optimizer
8:         Obtain the factorized components $\{A_t^i\}, \{B_t^j\}, \{\alpha_t^{i,j}\}$
9:     **end for**
10:     Copy the $l$-th layer down projection matrices $\{W_{\text{down}}^i\}_{i=1}^n$ from $\mathcal{M}_{\text{MoBE}}$
11:     Assemble the $l$-th MoBE layer with $\{A_t, B_t, \alpha_t\}$ and $\{W_{\text{down}}\}$.
12: **end for**
13: **return** $\mathcal{M}_{\text{MoBE}}$

---

We further analyze the parameter complexity of MoBE compared to standard MoE as illustrated in Table 1. Note that this analysis considers only the total and activation parameter count for a single MoE layer, excluding other components such as the embedding and attention layers. The total parameter counts for one MoBE layer is $ndp + 2npr + 2mrd$, where the first term is for the down matrices $W_{down}$, the second term is for the transformation matrices $A$ in the up and gate projection, and the third term is for the basis matrices $\{B^j\}$. The parameter count ratio ($\gamma$) from MoE to MoBE can be computed as

$$\gamma = \frac{ndp + 2npr + 2mrd}{3ndp} = \frac{1}{3} + \frac{2r}{3d} + \frac{2mr}{3np}.$$

Since $r \leq p < \frac{1}{2}d$, the second term $\frac{2r}{3d} < \frac{1}{3}$. For the last term, $m \ll n$, for an MoE with n = 128 experts, even if we set $m = 16$, we could have the last term $\frac{2mr}{3np} < \frac{1}{12}$. Therefore, $\gamma < \frac{1}{3} + \frac{1}{3} + \frac{1}{12} < 1$. When using MoBE to replace MoE, the compression ratio by MoBE is $1 - \gamma$. From the analysis, we can draw the conclusion that the MoBE architecture could substantially compress the standard MoE models.

Notably, while MoBE reduces the total parameters quite a lot, its activation parameter count requires closer examination. The matrices $\mathbf{B}$ and the down matrices $W_{down}$ contribute $2krd + kdp \leq 3kdp$ (since $r \leq p$) to the activation parameter count, while the transformation matrices $A$ introduce an additional $2kpr$. This may lead to an increase in the number of activation parameters. To compensate for this increase, inspired by previous work (Chaudhari et al., 2025), we propose a variant MoBE[†], which reduces the number of activated experts during inference from $k$ to a smaller value $k'$. In many modern MoE models, the number of activated experts $k$ is typically set to 8. In MoBE[†], we reduce this to 6 (i.e., $k' = 6$). *

## 3.3 ACTIVATION FUNCTION IN MOBE

In Eq(4), we employ an activation function $f$ to enhance representational power. However, not all activation functions are equally suitable. For instance, we posit that the commonly used ReLU (Glorot et al., 2011) activation function is suboptimal for this task. ReLU can induce excessive sparsity in the matrix $\mathbf{B}^i = f(\sum_{j=1}^m w^{i,j} B^j)$, which may cause notable information loss. As the transformation matrix $A^i \in \mathbb{R}^{p \times r}$ is smaller than $\mathbf{B}^i \in \mathbb{R}^{r \times d}$, it may struggle to compensate for this loss with such a limited representation capacity. Therefore, a bipolar activation function (i.e., one that outputs both positive and negative values like $\tanh$) is highly desirable.

Consequently, activation functions such as Tanh (LeCun et al., 1989), SiLU (Ramachandran et al., 2018), and GeLU (Hendrycks & Gimpel, 2016) are more suited for this task, while Sigmoid (Rumelhart et al., 1986) and ReLU are expected to yield inferior results. Our ablation study in Section 4.4 provide evidence supporting this hypothesis.

---

*The method (Chaudhari et al., 2025) reduces only activation parameters, not total parameters. Therefore, we consider it a complementary approach and did not include it in our experimental comparisons.

Table 1: Comparison of total and activation parameter count for one standard MoE and MoBE layer. MoBE$^\dagger$ is a MoBE variant with further activation expert number reduction.

|  | Standard MoE | MoBE | MoBE$^\dagger$ |
|---|---|---|---|
| #Total Parameters | $3ndp$ | $ndp + 2npr + 2mrd$ | $ndp + 2npr + 2mrd$ |
| #Activation Parameters | $3kdp$ | $kdp + 2kpr + 2krd$ | $k'dp + 2k'pr + 2k'rd$ |

Table 2: Means and stds of the gate matrices and up matrices in various MoE-based LLMs.

|  |  | Ling-Lite-Chat | DeepSeek-V2-Lite-Chat | DeepSeek-V3-0324 | Qwen3-30B-A3B-2507 | Qwen3-235B-A22B-2507 | Kimi-K2-Instruct |
|---|---|---|---|---|---|---|---|
| Gate Matrices | Mean | 2.2e-5 | 1.0e-6 | -4.2e-6 | -2.8e-5 | -1.4e-5 | -1.3e-6 |
|  | Std | 2.8e-2 | 2.9e-2 | 1.2e-2 | 2.3e-2 | 1.6e-2 | 2.6e-2 |
| Up Matrices | Mean | 2.3e-7 | -1.6e-7 | -5.3e-9 | 5.3e-7 | 1.8e-8 | 4.2e-8 |
|  | Std | 2.8e-2 | 3.0e-2 | 1.2e-2 | 2.3e-2 | 1.6e-2 | 2.6e-2 |

## 3.4 Z-score Normalization in MoBE

To address the impact of a wide range of weight values and obtain stable results in seeking the basis, we consider normalizing all expert weight matrices in each MoE layer. We introduce a Z-score normalization by subtracting the mean and dividing by the standard deviation (std) across all experts' weights:

$$\mu_W = mean(W^1, W^2, ..., W^n), \tag{6}$$

$$\sigma_W = std(W^1, W^2, ..., W^n), \tag{7}$$

$$W_Z^i = \frac{W^i - \mu_W}{\sigma_W}. \tag{8}$$

This normalization introduces additional inference overhead. After factorization, the $\sigma_W$ term can be folded into the transformation matrix $A^i$, and the $\mu_W$ term will require an extra bias operation during inference compared to the original form Eq(4).

$$\hat{W}^i = \sigma_W \hat{W}_Z^i + \mu_W = (\sigma_W A^i) f(\sum_{j=1}^m \alpha^{i,j} B^j) + \mu_W. \tag{9}$$

However, we empirically study different off-the-shelf MoE models and find that $\mu_W$ is typically negligibly small as shown in Table 2. We can therefore omit the term $\mu_W$ in Eq(9). That means, we only require absorbing $\sigma_W$ into $A^i$ without introducing extra parameters and computing overhead during inference.

## 4 Experiments

In this section, we evaluate the proposed MoBE approach on popular open-source MoE models and compare to state-of-the-art MoE compression methods (Section 4.3). We then conduct a set of ablation studies on activation functions (Section 4.4) and normalization schemes (Section 4.5).

### 4.1 Setup

**Models.** We evaluate our method, MoBE, on a suite of popular open-source MoE-based LLMs: Ling-Lite-Chat (Team et al., 2025b), DeepSeek-V2-Lite-Chat (Shao et al., 2024), DeepSeek-V3-0324 (Liu et al., 2024), Qwen3-30B-A3B-2507, Qwen3-235B-A22B-2507 (Yang et al., 2025) and Kimi-K2-Instruct (Team et al., 2025a).

**Baseline.** We compare our approach against two state-of-the-art MoE compression baselines, D$^2$-MoE (Gu et al., 2025) and MoLAE (Liu et al., 2025). Both MoBE and MoLAE are data-free compression methods, whereas D$^2$-MoE requires a calibration dataset, for which we use tulu-v3-sft-mixture (Lambert et al., 2024). Due to the high computational cost of its backward pass, applying D$^2$-MoE to very large models like Qwen3-235B-A22B-2507, DeepSeek-V3-0324 and Kimi-K2-Instruct is infeasible on a single 8xH100 GPU machine. Therefore, comparisons involving D$^2$-MoE

are excluded from these three larger models. In addition, we compared two additional baseline methods, MoNE Zhang et al. (2025) and Sub-MoE Li et al. (2025a), on Qwen3-30B-A3B-2507 and Qwen3-235B-A22B-2507, with the results presented in the Appendix F.

**Hyper-parameters.** Hyper-parameters are configured per case (models or methods). We provide a more detailed explanation in the Appendix E regarding the impact of the values of the number of basis matrices $m$ and the rank $r$.

- For Ling-Lite-Chat and DeepSeek-V2-Lite-Chat, MoBE uses $m = 4$ basis matrices and MoLAE uses 8 latent matrices. To compensate extra computing cost introduced by extra activation parameters in MoBE (Section 3.2), we reduce vctivated experts from $k = 6$ to $k' = 4$ in MoBE$^\dagger$.

- For Qwen3-30B-A3B-2507 and Qwen3-235B-A22B-2507, both MoBE and MoLAE use 32 basis/latent matrices. MoBE reduces activated experts from $k = 8$ to $k' = 6$ in MoBE$^\dagger$.

- For DeepSeek-V3-0324, both MoBE and MoLAE use 64 basis/latent matrices, with MoBE reducing $k$ from 8 to 6 in MoBE$^\dagger$.

- For Kimi-K2-Instruct, both MoBE and MoLAE use 128 basis/latent matrices, and MoBE similarly reduces $k$ from 8 to 6. Due to optimization challenges with 384 experts per layer, we split them into two groups, each trained with 64 basis matrices.

- For D$^2$-MoE, the rank of delta weights is set to 700 for Ling-Lite-Chat and DeepSeek-V2-Lite-Chat, and 420 for Qwen3-30B-A3B-2507.

- For simplicity, we set the rank $r = p$ in all our studies. It gets more compression ratio when setting $r < p$ while may increasing the accuracy drops.

**Implementation Details.** All experiments are conduct on H100 or H20 GPUs using the Adam optimizer (Loshchilov & Hutter, 2017) with a 0.07 learning rate. We set the batch size equal to the number of experts $n$ and train for a maximum of 50,000 epochs, employing early stopping with a patience of 2,000 epochs based on the training loss.

## 4.2 EVALUATION BENCHMARK

We perform a comprehensive evaluation across a wide spectrum of benchmark. The evaluation suite covers four primary domains: (1) **General Knowledge:** BBH (Srivastava et al., 2022), MMLU (Hendrycks et al., 2020), CEval (Huang et al., 2023), and CMMLU (Li et al., 2023a); (2) **General Reasoning:** ARC-Challenge (Clark et al., 2018), IFEval (Zhou et al., 2023), and GPQA (Rein et al., 2023); (3) **Mathematics:** Math (Hendrycks et al., 2021), GSM8k (Cobbe et al., 2021), AIME24, and AIME25; and (4) **Coding:** MBPP (Austin et al., 2021), HumanEval (Chen et al., 2021), LCB (LiveCodeBench-v5) (Jain et al., 2024), and MultiPL-E (Cassano et al., 2022). For **AIME24** and **AIME25**, we run 16 inference trials per question and report average accuracy; for **IFEval**, the final score is the mean of strict accuracies at both the prompt and instruction levels.

## 4.3 MAIN RESULTS

All the compared results of the origin model (MoE) and different compression methods (MoBE, MoBE$^\dagger$, D$^2$-MoE, and MoLAE) are shown in Tables 3. It shows that our proposed MoBE method generally outperforms all the compared compression methods across various benchmarks. For instance, for the Ling-Lite-Chat and DeepSeek-V2-Lite-Chat models, MoBE improves performance by 2-3% accuracy over the baseline. The performance gains are even more notable for Qwen3-30B-A3B-2507, Qwen3-235B-A22B-2507, DeepSeek-V3-0324 and Kimi-K2-Instruct, reaching 4-8% accuracy advantages over compared compression methods.

We note that converting MoE models into MoBE architecture results in an average performance degradation of 1.4% accuracy compared to the original MoE models. For comparison, MoBE$^\dagger$ that only reduces the number of activated experts from $k$ to $k'$, leads to a smaller degradation of 0.5% accuracy. It suggests that it is more challenging to compress the total parameters than activation parameters for an MoE model. As the sparsity ratio (#activated-parameters/#total-parameters) of recent MoE models becomes larger and larger so that the total parameter counts reach trillion-level ($\geq$1T), *it is more useful and practical to compression the total parameters*.

Table 3: Performance comparison of different compression methods on various MoE-based LLMs, where "†" indicates that this model activates fewer experts than the original model to compensate for the increase in activation parameters. The column *Ratio* refers to the proportion of compressed parameters to the total parameters in the LLMs.

| LLM | Method | Ratio | General Reasoning | | | General Knowledge | | | | Mathematics | | | | Coding | | | | Avg |
| --- | --- | --- | ARC-C | IFEval | GPQA | BBH | MMLU | CEval | CMMLU | Math | GSM8k | AIME24 | AIME25 | MBPP | HumanEval | Multipl-E | LCB | |
| Ling-Lite-Chat | MoE | 0% | 89.2 | 81.5 | 33.0 | 58.7 | 72.6 | 65.4 | 70.6 | 72.6 | 88.1 | 8.3 | 10.0 | 77.3 | 81.2 | 65.0 | 21.6 | 59.7 |
| | D²-MoE | 14% | 82.4 | 78.3 | **31.2** | 51.3 | 64.5 | 56.5 | 56.0 | 64.9 | 85.7 | 8.3 | 10.0 | 70.3 | 72.6 | 50.2 | 14.4 | 53.1 |
| | MoLAE | 12% | 85.4 | 75.1 | 29.7 | 51.9 | 69.5 | 61.9 | 62.3 | 65.3 | 83.9 | 10.0 | 4.2 | 71.4 | **82.9** | 60.3 | 15.0 | 55.3 |
| | MoBE | 16% | **87.1** | **79.2** | 29.4 | **61.5** | **71.5** | **66.6** | 66.2 | **70.4** | **88.0** | **11.7** | 9.2 | 77.5 | **82.9** | **64.0** | 14.4 | **58.6** |
| | MoBE† | 16% | 85.8 | **79.2** | 29.9 | 53.8 | 70.3 | 64.3 | **66.9** | 69.1 | 83.6 | **11.7** | **12.5** | 77.3 | 82.6 | 62.4 | **17.4** | 57.8 |
| DeepSeek-V2-Lite-Chat | MoE | 0% | 65.1 | 49.7 | 25.9 | 36.0 | 53.7 | 55.4 | 58.6 | 27.6 | 61.4 | 0 | 0 | 59.0 | 40.2 | 34.6 | 2.4 | 38.0 |
| | D²-MoE | 13% | 62.7 | **49.0** | 29.7 | 30.1 | 50.4 | 48.1 | 51.0 | 23.2 | **60.0** | 0 | 0 | 50.1 | 39.2 | 25.8 | 1.8 | 34.7 |
| | MoLAE | 11% | 65.8 | 43.9 | 26.9 | **34.0** | 53.0 | 47.9 | 52.8 | 18.6 | 59.2 | 0.8 | 0 | 46.6 | 41.5 | 26.9 | 1.8 | 34.6 |
| | MoBE | 15% | **67.5** | 46.0 | **30.3** | 33.9 | **53.7** | **53.0** | 56.3 | 23.5 | 58.6 | 0.8 | 0 | 51.5 | 43.3 | **31.7** | 3.6 | **36.9** |
| | MoBE† | 15% | 63.1 | 45.1 | 26.6 | 32.5 | 50.9 | **53.0** | 55.3 | 23.0 | **60.0** | **2.5** | 0 | 51.5 | **50.6** | 29.0 | 3.0 | 36.4 |
| Qwen3-30B-A3B-2507 | MoE | 0% | 95.6 | 86.6 | 56.8 | 85.4 | 87.6 | 88.2 | 86.6 | 93.3 | 96.4 | 59.4 | 51.3 | 86.4 | 93.1 | 70.6 | 41.5 | 78.6 |
| | D²-MoE | 24% | 93.1 | 83.5 | 45.2 | 69.9 | 83.3 | 71.2 | 68.6 | 86.1 | 93.0 | 38.3 | 29.1 | 79.5 | 84.0 | 44.0 | 26.9 | 66.4 |
| | MoLAE | 24% | 92.5 | 79.2 | 46.3 | 76.5 | 80.3 | 76.0 | 74.9 | 85.4 | 91.4 | 35.2 | 33.1 | 81.7 | 82.9 | 50.8 | 25.6 | 67.5 |
| | MoBE | 24% | **96.6** | **86.9** | **52.1** | **83.5** | 85.6 | 85.1 | 83.5 | 92.5 | 95.2 | **55.0** | 45.2 | **87.4** | 91.8 | **61.9** | 35.6 | **75.8** |
| | MoBE† | 24% | 95.9 | 85.1 | 51.0 | 83.3 | **86.0** | **85.9** | **83.9** | **92.6** | **96.1** | 54.0 | **45.6** | 85.3 | **92.2** | 61.2 | **38.0** | 75.7 |
| DeepSeek-V3-0324 | MoE | 0% | 97.0 | 84.8 | 66.7 | 85.4 | 90.3 | 90.4 | 88.6 | 92.0 | 94.9 | 56.9 | 47.3 | 89.7 | 93.4 | 68.2 | 44.6 | 79.3 |
| | MoLAE | 30% | 97.3 | 83.2 | 54.0 | 82.9 | 87.3 | 84.4 | 83.2 | 87.6 | **95.5** | 38.5 | 29.6 | 87.4 | 89.5 | 61.0 | 34.4 | 73.1 |
| | MoBE | 30% | **98.0** | **84.5** | **63.6** | 85.2 | **89.5** | 87.8 | 87.2 | 90.3 | 93.7 | **52.3** | 40.6 | **89.9** | 93.6 | **73.1** | 40.9 | **78.0** |
| | MoBE† | 30% | 96.6 | 84.3 | 62.6 | **85.4** | 87.2 | **87.9** | **89.4** | **91.0** | 94.8 | 49.8 | **41.9** | 89.0 | **93.8** | 73.0 | **42.1** | 77.9 |
| Qwen3-235B-A22B-2507 | MoE | 0% | 97.0 | 90.0 | 60.7 | 89.5 | 90.9 | 90.9 | 90.0 | 94.4 | 96.7 | 61.9 | 51.7 | 93.0 | 96.3 | 70.5 | 48.4 | 81.5 |
| | MoLAE | 24% | 95.6 | 85.5 | **66.2** | 87.5 | 88.9 | 87.3 | 86.9 | 90.5 | 95.5 | 54.2 | 44.6 | 70.7 | 81.0 | 30.0 | 33.5 | 73.2 |
| | MoBE | 24% | **96.3** | **89.9** | 58.6 | **89.0** | **90.4** | **90.6** | **89.7** | **94.2** | 96.3 | **64.8** | **54.8** | **89.2** | **93.8** | **71.9** | 43.7 | **80.9** |
| | MoBE† | 24% | 95.6 | 88.7 | 58.1 | 88.8 | 90.3 | 90.4 | 89.6 | 93.6 | 96.0 | 62.9 | 50.8 | 87.4 | 93.1 | 65.5 | **45.2** | 79.7 |
| Kimi-K2-Instruct | MoE | 0% | 95.9 | 90.8 | 77.4 | 88.8 | 90.8 | 92.4 | 89.9 | 95.7 | 96.7 | 64.8 | 50.2 | 90.9 | 95.4 | 66.7 | 50.3 | 82.4 |
| | MoLAE | 24% | 96.6 | 88.2 | 66.4 | 86.0 | 89.2 | 89.4 | 87.8 | 90.9 | 93.0 | 44.8 | 33.0 | 88.8 | 91.9 | 60.6 | 40.3 | 76.6 |
| | MoBE | 24% | **97.0** | 91.4 | 73.2 | 87.2 | **90.3** | 90.2 | 89.2 | 94.9 | 96.3 | **62.5** | **44.4** | 89.9 | 94.0 | **68.8** | **47.2** | **81.1** |
| | MoBE† | 24% | 96.3 | **91.7** | **74.6** | **88.1** | 90.2 | **90.3** | **89.3** | **95.1** | **96.6** | 61.7 | 44.2 | **90.4** | **94.1** | 65.0 | 44.6 | 80.8 |

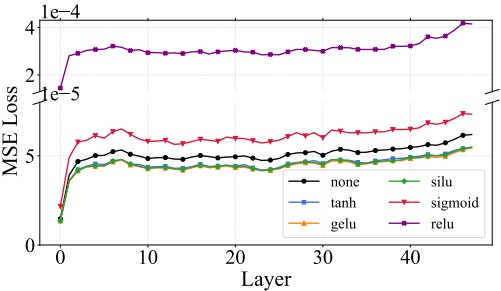

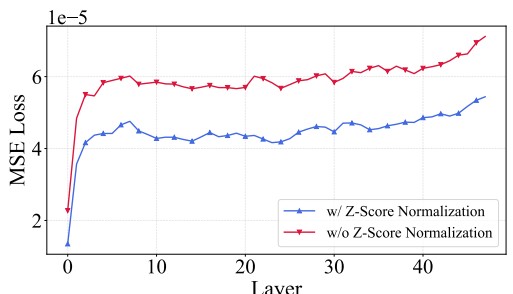

Figure 4: Comparison of per-layer MSE loss for compressing the gate matrices of Qwen3-30B-A3B when using different activation functions.

Figure 5: Comparison of per-layer MSE loss for compressing the gate matrices of Qwen3-30B-A3B with/without Z-score normalization.

## 4.4 ABLATION STUDY ON ACTIVATION FUNCTIONS

In Eq(4), we apply a non-linear activation function to enhance representational capacity. We conduct experiments on the Qwen3-30B-A3B model's gate matrices to select the optimal activation function. As shown in Figure 4, Sigmoid demonstrates inferior performance to the case without activation in terms of the reconstruction MSE, while ReLU has an order-of-magnitude higher MSE loss. This result is consistent with our analysis in Section 3.3. GELU, SiLU, and Tanh achieve similar results and outperform the case without activation, while we finally choose SiLU and Tanh as our activation function as they offer a favorable trade-off between performance and computational efficiency.

## 4.5 ABLATION STUDY ON Z-SCORE NORMALIZATION

To evaluate the impact of the Z-score normalization introduced in Section 3.4, we conduct an ablation study using the Qwen3-30B-A3B's gate matrices. All experiments use identical hyperparameter and optimization settings, varying only the application of normalization. Figure 5 shows a notable reduction in MSE loss when Z-score normalization is applied. We hypothesize that the normalization can rescale the weight values from wide and wild ranges to a normal distribution with a mean of 0 and a std of 1, so that the optimization becomes more stable and effective.

## 5 CONCLUSION

In this paper, we propose the Mixture-of-Basis-Experts (MoBE), a parameter-efficient architecture that addresses memory challenges in deploying large-scale MoE-based LLMs. MoBE effectively combines shared basis matrices with expert-specific transformation matrices via rank decomposition to overcome limitations of prior work. Extensive experiments demonstrate that MoBE outperforms existing counterpart methods like MoLAE and $D^2$-MoE with a large margin in preserving higher performance and a better model compression rate. MoBE can compress leading models such as Qwen3-235B-A22B-2507, DeepSeek-V3-0324 and Kimi-K2-Instruct by up to 24%-30% while retaining up to 98% of their original performance across diverse benchmarks. Such a practical and effective method may help enable large MoE models for more scalable and efficient applications.

## STATEMENT ON THE USE OF LARGE LANGUAGE MODELS

In preparing this manuscript, Large Language Models were used exclusively for refining language, grammar, and clarity. The core ideas and content remain entirely the author(s)' own, who bear full responsibility for all information presented herein.

## LIMITATIONS

While our method performs well in compressing MoE models, it still causes a slight drop in accuracy compared to the original model. To fix this gap, one potential direction is to employ full network knowledge distillation (KD) between the original and our compressed models. This requires modifying existing training frameworks to support KD training for large LLMs. Another limitation is that MoBE requires multiple times calling of current optimized kernel fused-MoE to mimic the factorization, which is relatively inefficient. Hence, it requires implementing a specific mega-kernel for the whole factorization to unleash the power of the MoBE architecture. Future work will address these two limitations.

## ACKNOWLEDGMENTS

This work is supported by the National Key Research & Development Plan (2023YFF0725100) and the National Natural Science Foundation of China (92570121, 62322214, U23A20299, U24B20144). We also acknowledge the support of the Ant Group Research Intern Program.

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

# A ABSOLUTE PERFORMANCE COMPARISON OF MoE COMPRESSION METHODS

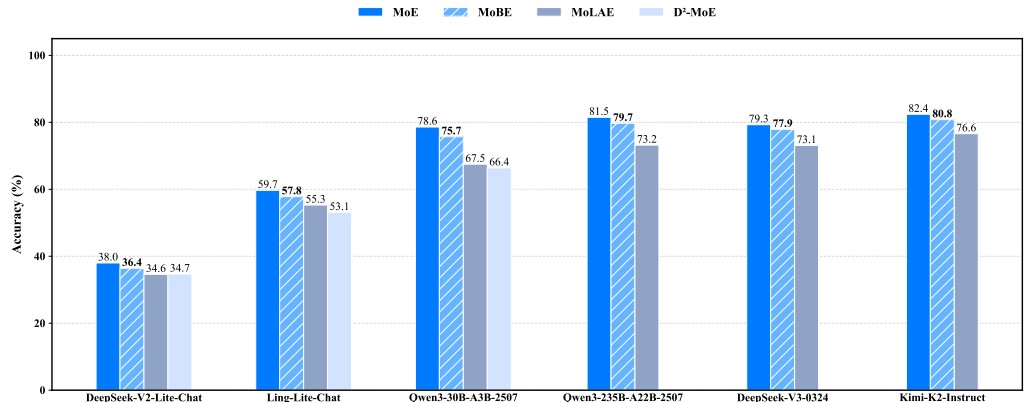

Figure 6: Absolute performance comparison of different MoE compression methods.

We present the absolute performance comparison of MoE compression methods in Figure 6.

# B ANALYSIS OF THE EFFECTIVE RANK OF EXPERT WEIGHT MATRICES

We evaluate the effective rank of expert weight matrices in Qwen3-235B-A22B-2507, DeepSeek-V3-0324, and Kimi-K2-Instruct. The effective rank $r_e$ is defined as:

$$r_e = \min\left\{ k \in \mathbb{N}^+ \,\middle|\, \frac{\sum_{j=1}^{k} \sigma_j^2}{\sum_{i=1}^{r} \sigma_i^2} > 0.95 \right\}$$

where $\sigma_i$ is the $i$-th largest singular value (sorted in descending order) and $r$ is the matrix rank. The expert weight matrices in Qwen3-235B-A22B-2507 have dimensions 4096×1536, while those in DeepSeek-V3-0324 and Kimi-K2-Instruct are 7168×2048. Figures 8–10 illustrate the per-layer average effective rank $\overline{r_e}$ and its range for each model. Taking the expert weight matrices of Kimi-K2-Instruct as an example, rank decomposition could realize parameter compression only if the intermediate rank satisfies

$$r_t \leq \frac{7168 \cdot 2048}{7168 + 2048} \approx 1593.$$

However, according to Figure 10, the average effective rank $\overline{r_e}$ is larger than 1593 in most layers. This discrepancy implies that the pure rank-decomposition-based method can't produce model compression without performance loss. An interesting finding can be drawn from the analysis: Qwen3-235B-A22B-2507 shows much broader effective rank range than the other two, which may indicate that its experts are far from being well-balanced during the training phase.

# C ADDITIONAL MSE COMPARISONS

We present a comparison of reconstruction errors on Ling-Lite-Chat, DeepSeek-V2-Lite, Qwen3-235B-A22B-2507, DeepSeek-V3-0324 and Kimi-K2-Instruct in the Figure 11-15.

# D AN EXPRESSIVE POWER ANALYSIS OF MoBE

In this section, we demonstrate that the MoBE possesses greater expressive capacity than a conventional low-rank factorization derived from Singular Value Decomposition (SVD). Assume there are $n$ experts in one layer, each with an up/gate matrix of dimension $p \times d$. To construct the SVD-based baseline, we first partition these $n$ experts into $m$ groups, with each group containing $\frac{n}{m}$ matrices (assuming $n$ is a multiple of $m$). For the $j$-th group, the up/gate matrices are stacked row-wise to

form a consolidated matrix $W_{\text{stack}}^j \in \mathbb{R}^{\frac{np}{m} \times d}$. We perform Singular Value Decomposition (SVD) on this stacked matrix:

$$U^j, \Sigma^j, V^j = \text{SVD}(W_{\text{stack}}^j) \tag{10}$$

where $U^j \in \mathbb{R}^{\frac{np}{m} \times \frac{np}{m}}$, $\Sigma^j \in \mathbb{R}^{\frac{np}{m} \times d}$, and $V^j \in \mathbb{R}^{d \times d}$. The decomposition is truncated to retain the top $r$ singular values, obtaining the components $\tilde{U}^j \in \mathbb{R}^{\frac{np}{m} \times r}$, $\tilde{\Sigma}^j \in \mathbb{R}^{r \times r}$, and $\tilde{V}^j \in \mathbb{R}^{r \times d}$. We then incorporate $\tilde{\Sigma}^j$ into $\tilde{U}^j$ and $\tilde{V}^j$ by setting $\tilde{U}^j = \tilde{U}^j(\tilde{\Sigma}^j)^{1/2}$ and $\tilde{V}^j = (\tilde{\Sigma}^j)^{1/2}\tilde{V}^j$. For the $i$-th up/gate matrix, assuming it belongs to the $j$-th group, its low-rank approximation is given by:

$$\hat{W}^i = A^i B^j \tag{11}$$

where $A^i$ corresponds to $\tilde{U}^j[(k-1)p : kp, :]$ with $k = i - (j-1)(\frac{n}{m})$, indicating that the $i$-th up/gate matrix is the $k$-th matrix in the $j$-th group, and $B^j$ is $\tilde{V}^j$. Comparing this factorization with Eq(4), we observe that the SVD-based decomposition is a special case of MoBE. Specifically, it is equivalent to MoBE where the $\alpha$ are restricted to one-hot vectors (assigning each expert to a single group) and the activation function is an identity mapping. Therefore, MoBE exhibits significantly greater expressive power than this SVD-based approach.

## E  ANALYSIS OF BASIS MATRIX COUNT AND RANK

The number of parameters for the basis matrices is $mrd$, where $m$ is the number of basis matrices, $r$ is the rank of the basis matrices, and $d$ is the hidden dimension of the model. We conduct experiments on Qwen3-30B-A3B-2507 with different values of $m$ and $r$; a comparison of the reconstruction errors is presented in Figure 7.

As shown in the Figure 7, for a fixed parameter budget, the reconstruction error for the configuration $m = 32, r = 768$ is significantly lower than that for $m = 64, r = 384$. This indicates that the rank $r$ of the basis matrices is a more influential factor than the number of basis matrices $m$. Therefore, in our main experiments, we set $r$ to be equal to the hidden dimension of the MoE to maximize representational capacity.

Additionally, we conduct experiments on Qwen3-30B-A3B-2507 with $r$ fixed at 768 while varying $m$ to investigate its impact on downstream task performance. The results are presented in Table 4. These results show that when $m = 16$, the performance of the compressed model degrades substantially compared to when $m = 32$. Conversely, while performance at $m = 64$ is marginally better than at $m = 32$, this improvement comes at the cost of a considerably lower compression rate. Therefore, in our main experiments, we select configurations corresponding to a 25%–30% compression rate to achieve a favorable balance between compression efficiency and model performance.

Table 4: Performance comparison of different configuration settings on Qwen3-30B-A3B-2507. The column *Ratio* refers to the proportion of compressed parameters to the total parameters in the LLMs.

| LLM | m | Ratio | General Reasoning | | | | General Knowledge | | | | Mathematics | | | | Coding | | | | Avg |
| --- | --- | --- | --- | --- | --- | --- | --- | --- | --- | --- | --- | --- | --- | --- | --- | --- | --- | --- | --- |
| | | | ARC-C | IFEval | GPQA | BBH | MMLU | CEval | CMMLU | Math | GSM8k | AIME24 | AIME25 | MBPP | HumanEval | Multipl-E | LCB | |
| Qwen3-30B-A3B-2507 | 16 | 32% | 93.5 | 80.6 | 48.2 | 78.3 | 82.3 | 79.2 | 78.9 | 88.1 | 93.2 | 40.6 | 37.1 | 83.8 | 86.0 | 55.3 | 27.6 | 70.2 |
| | 32 | 24% | 96.6 | 86.9 | 52.1 | 83.5 | 85.6 | 85.1 | 83.5 | 92.5 | 95.2 | 55.0 | 45.2 | 87.4 | 91.8 | 61.9 | 35.6 | 75.8 |
| | 64 | 8% | 94.5 | 85.5 | 53.8 | 84.8 | 86.6 | 87.0 | 85.3 | 92.2 | 96.4 | 57.1 | 48.3 | 86.0 | 92.7 | 66.2 | 38.4 | 77.0 |

## F  ADDITIONAL BASELINE COMPARISONS

We compared two additional baseline methods on Qwen3-30B-A3B-2507 and Qwen3-235B-A22B-2507: MoNE and Sub-MoE. MoNE prunes less important experts based on their routing weights and output variance, replacing them with their average outputs. Sub-MoE groups experts by output cosine similarity and merges each group into one. This merge uses SVD on the expert matrices, weighting their components by activation frequency to create a new, single expert. For both of these methods, we use tulu-v3-sft-mixture as the calibration dataset. As shown in the Table 5, MoBE consistently outperforms these baseline methods.

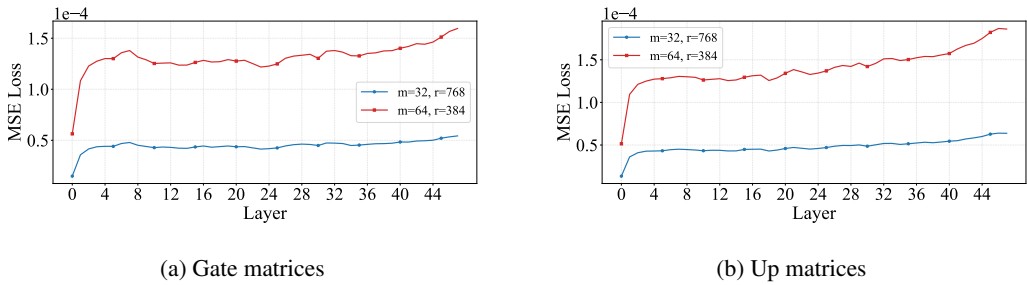

(a) Gate matrices
(b) Up matrices

Figure 7: Comparison of pre-layer MSE for compressing the gate (a) and up (b) matrices of Qwen3-30B-A3B-2507 using different configuration settings.

Table 5: Performance comparison of different compression methods on various MoE-based LLMs, where "†" indicates that this model activates fewer experts than the original model to compensate for the increase in activation parameters. The column *Ratio* refers to the proportion of compressed parameters to the total parameters in the LLMs.

| LLM | Method | Ratio | General Reasoning | | | General Knowledge | | | | Mathematics | | | | Coding | | | | Avg |
|---|---|---|---|---|---|---|---|---|---|---|---|---|---|---|---|---|---|---|
| | | | ARC-C | IFEval | GPQA | BBH | MMLU | CEval | CMMLU | Math | GSM8k | AIME24 | AIME25 | MBPP | HumanEval | Multipl-E | LCB | |
| Qwen3-30B-A3B-2507 | MoE | 0% | 95.6 | 86.6 | 56.8 | 85.4 | 87.6 | 88.2 | 86.6 | 93.3 | 96.4 | 59.4 | 51.3 | 86.4 | 93.1 | 70.6 | 41.5 | 78.6 |
| | D²-MoE | 24% | 93.1 | 83.5 | 45.2 | 69.9 | 83.3 | 71.2 | 68.6 | 86.1 | 93.0 | 38.3 | 29.1 | 79.5 | 84.0 | 44.0 | 26.9 | 66.4 |
| | MoLAE | 24% | 92.5 | 79.2 | 46.3 | 76.5 | 80.3 | 76.0 | 74.9 | 85.4 | 91.4 | 35.2 | 33.1 | 81.7 | 82.9 | 50.8 | 25.6 | 67.5 |
| | MoNE | 24% | 92.5 | 85.1 | 52.0 | 80.7 | 80.1 | 71.2 | 68.2 | 92.6 | 94.8 | 53.2 | 44.5 | 85.6 | 91.8 | 57.8 | 38.0 | 72.5 |
| | Sub-MoE | 24% | 92.5 | 82.1 | 46.3 | 82.8 | 80.3 | 70.2 | 68.5 | 92.0 | 96.0 | 51.0 | 45.2 | 82.2 | 87.2 | 50.1 | 34.1 | 70.7 |
| | MoBE | 24% | 96.6 | 86.9 | 52.1 | 83.5 | 85.6 | 85.1 | 83.5 | 92.5 | 95.2 | 55.0 | 45.2 | 87.4 | 91.8 | 61.9 | 35.6 | 75.8 |
| | MoBE† | 24% | 95.9 | 85.1 | 51.0 | 83.3 | 86.0 | 85.9 | 83.9 | 92.6 | 96.1 | 54.0 | 45.6 | 85.3 | 92.2 | 61.2 | 38.0 | 75.7 |
| Qwen3-235B-A22B-2507 | MoE | 0% | 97.0 | 90.0 | 60.7 | 89.5 | 90.9 | 90.9 | 90.0 | 94.4 | 96.7 | 61.9 | 51.7 | 93.0 | 96.3 | 70.5 | 48.4 | 81.5 |
| | MoLAE | 24% | 95.6 | 85.5 | 66.2 | 87.5 | 88.9 | 87.3 | 86.9 | 90.5 | 95.5 | 54.2 | 44.6 | 70.7 | 81.0 | 30.0 | 33.5 | 73.2 |
| | MoNE | 24% | 94.0 | 88.3 | 58.7 | 86.6 | 87.2 | 82.3 | 78.1 | 93.0 | 95.9 | 62.9 | 50.4 | 87.8 | 93.3 | 60.3 | 44.3 | 77.5 |
| | Sub-MoE | 24% | 93.8 | 85.8 | 50.4 | 87.8 | 87.3 | 81.2 | 80.8 | 92.5 | 95.5 | 61.6 | 50.4 | 83.4 | 89.0 | 58.8 | 42.2 | 76.0 |
| | MoBE | 24% | 96.3 | 89.9 | 58.6 | 89.0 | 90.4 | 90.6 | 89.7 | 94.2 | 96.3 | 64.8 | 54.8 | 89.2 | 93.8 | 71.9 | 43.7 | 80.9 |
| | MoBE† | 24% | 95.6 | 88.7 | 58.1 | 88.8 | 90.3 | 90.4 | 89.6 | 93.6 | 96.0 | 62.9 | 50.8 | 87.4 | 93.1 | 65.5 | 45.2 | 79.7 |

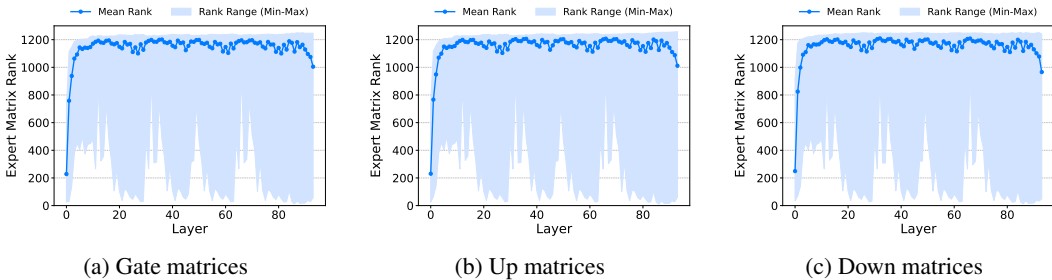

(a) Gate matrices
(b) Up matrices
(c) Down matrices

Figure 8: Average effective rank and effective rank range of the (a) gate, (b) up, and (c) down matrices at each layer in Qwen3-235B-A22B-2507.

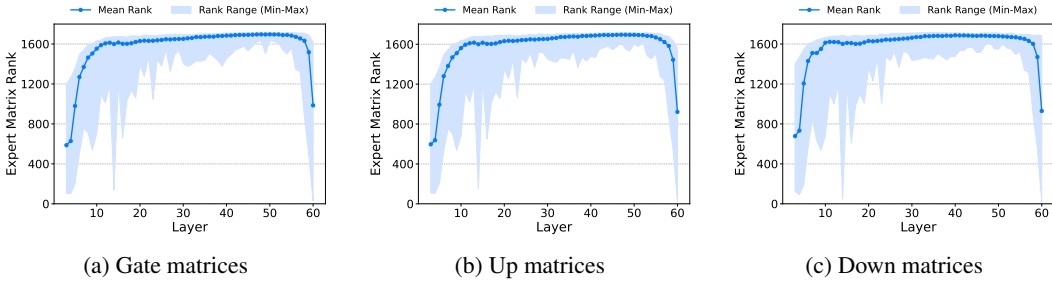

(a) Gate matrices
(b) Up matrices
(c) Down matrices

Figure 9: Average effective rank and effective rank range of the (a) gate, (b) up, and (c) down matrices at each layer in DeepSeek-V3-0324.

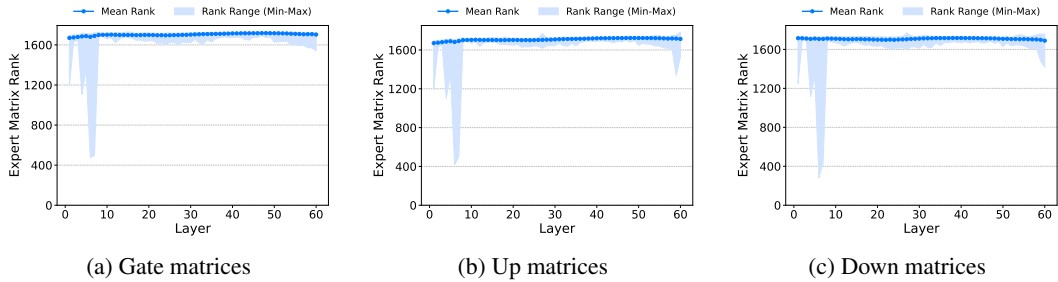

(a) Gate matrices      (b) Up matrices      (c) Down matrices

Figure 10: Average effective rank and effective rank range of the (a) gate, (b) up, and (c) down matrices at each layer in Kimi-K2-Instruct.

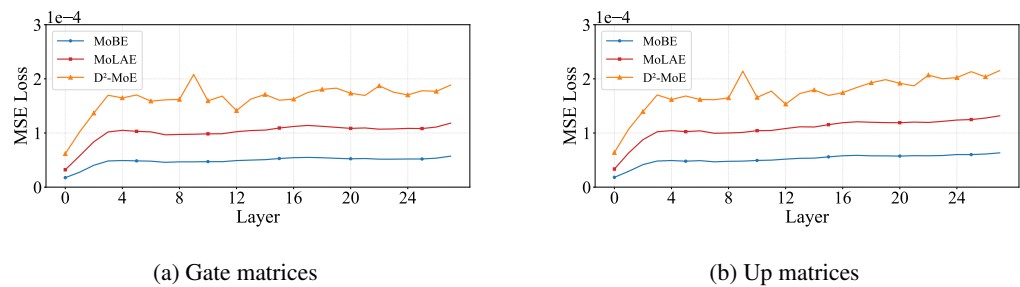

(a) Gate matrices              (b) Up matrices

Figure 11: Comparison of pre-layer MSE for compressing the gate (a) and up (b) matrices of Ling-Lite-Chat using MoBE, $D^2$-MoE and MoLAE.

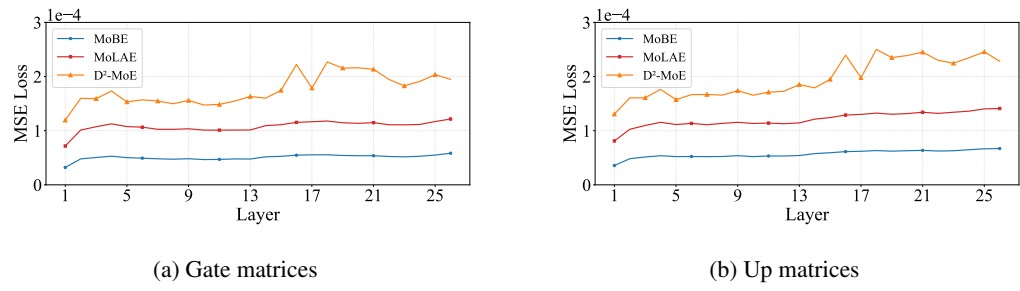

(a) Gate matrices              (b) Up matrices

Figure 12: Comparison of pre-layer MSE for compressing the gate (a) and up (b) matrices of DeepSeek-V2-Lite-Chat using MoBE, $D^2$-MoE and MoLAE.

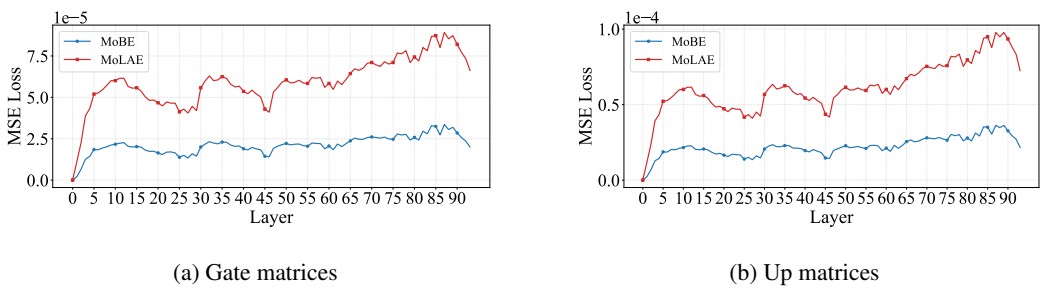

(a) Gate matrices              (b) Up matrices

Figure 13: Comparison of pre-layer MSE for compressing the gate (a) and up (b) matrices of Qwen3-235B-A22B-2507 using MoBE and MoLAE.

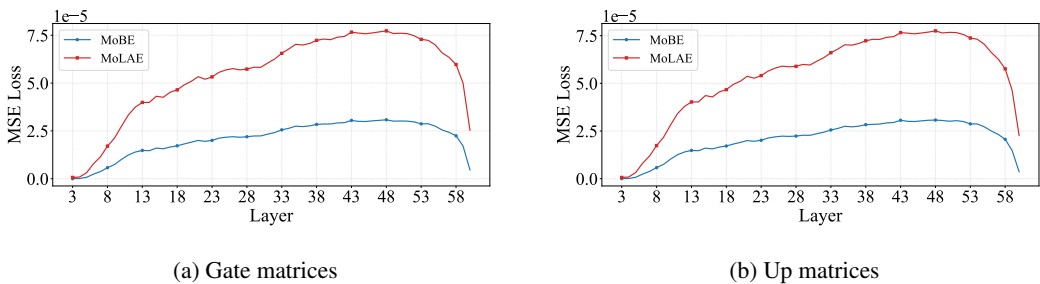

(a) Gate matrices

(b) Up matrices

Figure 14: Comparison of pre-layer MSE for compressing the gate (a) and up (b) matrices of DeepSeek-V3-0324 using MoBE and MoLAE.

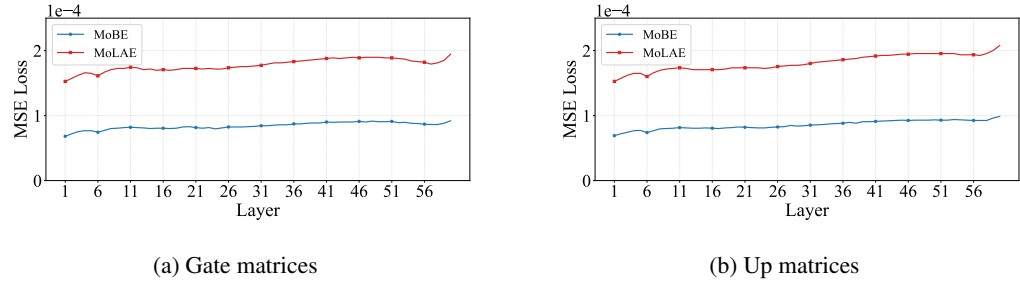

(a) Gate matrices

(b) Up matrices

Figure 15: Comparison of pre-layer MSE for compressing the gate (a) and up (b) matrices of Kimi-K2-Instruct using MoBE and MoLAE.

