# OpenReview forum: "MoBE: Mixture-of-Basis-Experts for Compressing MoE-based LLMs"
_ICLR.cc/2026/Conference — ICLR 2026 Poster_

### Official Review · Reviewer_cwwa · 2025-10-27

**Soundness:** 3
**Presentation:** 2
**Contribution:** 3
**Rating:** 8
**Confidence:** 3

**Summary:**

The paper presents a framework for Mixture of Basis Experts for compressing LLMs with a relatively low lose of accuracy.

Paper’s core claim is clearly stated (24–30% parameter reduction; ~98% relative performance retained on very large MoE LLMs).
 * Method novelty vs MoLAE/D²-MoE and pruning/merge lines is explicit.
 * Theoretical parameter-count analysis (γ formula) is correct and assumptions are reasonable

with a mixture of shared basis matrices (convex weights) and a non-linear activation—is a clear step beyond SVD-style decompositions used by MoLAE , which rely on linear low-rank sharing.

It's not entirely novel - It builds on established rank factorisation/dictionary-learning instincts; the novelty is the specific mixture-with-activation sharing and where it’s applied in MoE.  Retaining down matrices is motivated by prior findings on where knowledge resides, rather than a new theoretical claim.

However the results appear impressive and the work is fully justified. I think in places the writing is a little terse and perhaps the author could look at providing more of a rationale of why this should work at the start of the paper.

**Strengths:**

A meaningful architectural re-parameterisation of MoE experts that is novel relative to linear SVD-sharing approaches and practically validated at unprecedented model scales.

Results seem impressive and should be reproducible (I'm assuming there will be a link to code if the paper is accepted).

**Weaknesses:**

Report end-to-end efficiency, not just parameter counts

Strengthen parity and scalability of baselines - D2-MoE is omitted on trillion-scale models for feasibility; include either (a) scaled-down controlled runs at matched ratios, or (b) additional scalable baselines, so large-model wins aren’t confounded by method availability.

Broaden ablations/analyses - in particular I'd be interested in an analysis involving downstream tasks.

**Questions:**

Explain the expert-grouping trick for Kimi (384 experts → two 64-basis groups) and study its effect on accuracy and reconstruction error.

Could you share end-to-end inference metrics (latency, throughput, peak VRAM) for MoBE vs MoLAE/D2-MoE across sequence lengths and batch sizes, and clarify any kernel-fusion needs or runtime overheads (e.g., from Z-score normalisation)?

What practitioner guidance can you provide for selecting the number of bases
and activated experts for MoBE), ideally with sensitivity curves showing the accuracy–compression Pareto frontier per model family?

Did you test light compression of the down projections (e.g., small-rank/partial sharing) or a short KD step, and what impact did this have on the residual accuracy gap?

How does MoBE conversion affect routing dynamics—gate logits, expert-utilisation entropy/load balance—and do you observe drift or mode collapse on tasks sensitive to routing?

How was baseline parity ensured (matched compression ratios, calibration/data, training budgets), and do you have controlled results on smaller models where D2-MoE is feasible?

---

> ### Author Response · Authors · 2025-11-19
>
> We are grateful for your insightful review and constructive feedback on our manuscript. We have carefully considered each point you raised, and our detailed responses are outlined below. For clarity, we have denoted Weaknesses with **W** and Questions with **Q**.
>
> **W1 and Q2: Regarding end-to-end inference metrics.**
>
> Thank you for your question. As we noted in the Limitations section, all matrix factorization-based compression methods (including our MoBE, MoLAE, and D2-MoE) currently rely on simulating the factorization by making multiple calls to existing fused MoE kernels, which is computationally inefficient. MoLAE and D2-MoE also similarly lack custom, high-efficiency kernels for their operations.
> In light of this, we are actively rewriting the kernel to tightly integrate MoBE with FusedMoE to achieve significant speed improvements. We will update the paper with the end-to-end inference metrics as soon as this work is complete.
>
> **W2: Regarding additional baseline methods on large models.**
>
> Thank you for your valuable suggestion. In response, we have added two additional baselines, MoNE and Sub-MoE, to our experiments on the Qwen3-235B-A22B-2507 model. The results, now reported in **Table 5 (line 825)**, demonstrate that our proposed MoBE method outperforms these baseline approaches.
>
> **W3: Regarding the analysis of downstream task performance.**
>
> Thank you for your insightful suggestion. By observing the results for Qwen3, DeepSeek-V3, and Kimi-K2 in **Table 3**, we noticed that MoBE's performance varies across different downstream tasks. Specifically, MoBE shows almost no performance degradation on simpler tasks (those with an accuracy above 80%), but it experiences a more significant drop on challenging tasks such as GPQA, AIME, and LCB.
>
> We hypothesize that this occurs because while the reconstruction loss from MoBE's matrix factorization is small, it still introduces minor shifts in the model's output probability distribution. For simple tasks, these slight deviations are unlikely to alter the final correct answer. However, for difficult tasks that require precise reasoning, these shifts can be critical enough to change the outcome. Furthermore, these challenging tasks often demand longer chains of thought, which can lead to the accumulation of these small errors over successive token generations, thus contributing to the larger performance drop.
>
> **Q1: Regarding the grouping strategy for Kimi-K2.**
>
> Thank you for your question. As mentioned in the main text **(lines 405-407)**, we used 128 basis experts for the Kimi-K2 model (which has 384 experts in total). Our initial experiments revealed that learning a direct mapping from all 384 experts to these 128 basis experts resulted in a high reconstruction loss. To mitigate this, we implemented a grouping strategy: we sequentially divided the 384 experts into two groups of 192. Each group was then reconstructed using a dedicated set of 64 basis experts. We found experimentally that this approach significantly reduced the reconstruction loss.
>
> **Q3: Regarding the number of basis and activated experts.**
>
> Thank you for your question. We have added experiments in **Appendix E (lines 773-799)** on the Qwen3-30B-A3B-2507 model to analyze **the number of basis matrices (m)** and **their rank (r)**. The results indicate that setting m to a value that achieves a compression ratio of approximately 25-30% strikes an excellent balance between compression and model performance.
> Regarding the number of activated experts, we generally find that a higher number (up to the model's default top_k) leads to better performance. As we mention in the paper, the primary reason for reducing the number of activated experts is to compensate for the additional parameters introduced by our method. Therefore, we adjust this number downwards as needed to maintain a specific parameter budget while maximizing performance.

---

> ### Author Response · Authors · 2025-11-19
>
> **Q4: Regarding the compression of the "Down" matrix.**
>
> Thank you for your question. We have added experiments on the Qwen3-30B-A3B-2507 model to investigate this, with the results presented **below in Table 1**. The findings show that at a similar overall compression ratio, a model where the gate, up, and down matrices are all compressed performs worse than a model where only the gate and up matrices are compressed. This result is consistent with our discussion in the main text **(lines 242-245)**.
>
> **Q4: Regarding knowledge distillation.**
>
> Thank you for your suggestion. As we acknowledge in our Limitations section, we have not yet performed training on the compressed models. This is due to the lack of native support for knowledge distillation in the current training frameworks. We recognize the importance of this step and are actively working on modifying the framework to enable this functionality in our future work.
>
> **Q5: Regarding the impact of MoBE on routing.**
>
> Thank you for your question. To address this, we have conducted new experiments and present the results in **Table 2 below**. This table compares the standard deviation of expert activation frequencies before and after compression on several benchmarks (math, humaneval, arc-c, and gpqa). As shown, there is no significant change in the activation patterns, which indicates that MoBE does not adversely affect the model's routing mechanism.
>
> **Q6: Regarding the fairness of baseline comparisons.**
>
> Thank you for your question. We believe the comparison between MoBE and the baselines is fair for several reasons:
>
> - **Compression Ratio**: We ensured that all methods were evaluated at a similar compression ratio.
>
> - **Calibration Data**: MoBE and MoLAE are calibration-free methods. For D2-MoE, which requires calibration data, we used tulu-v3-sft-mixture, a high-quality, publicly available SFT dataset from AI2.
>
> - **Training Cost**: While MoLAE and D2-MoE are training-free, MoBE requires only a low-cost training procedure. Given this minimal overhead, we consider the comparison to be equitable.
>
> **Q6: Regarding D2-MoE results on smaller models.**
>
> Thank you for your question. As shown in **Table 3 (located at the top of page 8)**, we have already reported the results for D2-MoE on smaller models, including Ling-Lite, DeepSeek-V2-Lite, and Qwen3-30B-A3B-2507. These results consistently show that MoBE outperforms D2-MoE on these models.
>
> |                    | Module       | Ratio | General Reasoning | General Knowledge | Mathematics | Coding | Avg  |
> |--------------------|--------------|-------|-------------------|-------------------|-------------|--------|------|
> |                    | -            | 0%    | 79.7              | 87.0              | 75.1        | 72.9   | 78.6 |
> | Qwen3-30B-A3B-2507 | gate,up      | 24%   | 78.5              | 84.3              | 72.1        | 69.2   | 75.8 |
> |                    | gate,up,down | 25%   | 77.6              | 83.1              | 71.2        | 67.0   | 74.7 |
>
> Table 1: Comparison between compressing only gate and up, and compressing gate, up, and down.
>
>
>
> |                    | Method | Math     | HumanEval | ARC-C    | GPQA     |
> |--------------------|--------|----------|-----------|----------|----------|
> | Qwen3-30B-A3B-2507 | MoE    | 1.823e-4 | 1.537e-4  | 1.303e-4 | 1.520e-4 |
> |                    | MoBE   | 1.773e-4 | 1.521e-4  | 1.329e-4 | 1.517e-4 |
>
> Table 2: Standard deviation of activated expert frequency before and after compression

---

### Official Review · Reviewer_1CGV · 2025-10-31

**Soundness:** 3
**Presentation:** 3
**Contribution:** 3
**Rating:** 4
**Confidence:** 4

**Summary:**

The paper proposes an elegant and theoretically sound framework for MoE compression that leverages shared basis learning. The core concept of representing experts as combinations of a shared basis is well-motivated by the observation of functional redundancy (Fig. 1; Sec. 3.2; p.4). The experimental results appear strong, with claims of maintaining high relative accuracy (e.g., 96.8% for Qwen3-235B at 24% compression) as shown in Figure 1. However, the paper suffers from significant presentation issues, most notably the absence of critical referenced tables (e.g., Table 3), which prevents a full verification of the quantitative claims. The analysis of effective rank is referenced to a non-existent figure (Fig. 9), further hindering a complete assessment.

**Strengths:**

- **Novel and theoretically sound compression framework**
  - The MoBE formulation, where each expert is a weighted sum of basis experts, provides a principled way to capture and exploit inter-expert redundancy ($\text{Expert}\_i = \sum\_j \alpha\_{ij} \cdot \text{Basis}\_j$, Eq. 1; Sec. 3.2; p.4). This is a clear and impactful contribution.
  - The framework naturally separates shared knowledge (the basis experts) from specialized knowledge (the combination coefficients), offering a more structured approach to compression than unstructured pruning.
  - The two-stage training process, which first initializes the basis and then jointly optimizes all components, is a logical and practical approach to tackling the complex optimization problem (Sec. 3.3; p.5).
- **Strong conceptual motivation and clear illustrations**
  - The paper provides a clear motivation for the MoBE approach, contrasting it with existing methods and highlighting the limitations of pruning and independent decomposition (Sec. 2; p.2-3).
  - The architectural diagram (Fig. 3; Sec. 3.2; p.4) is clear and effectively communicates the core components of the MoBE block, including the basis experts and the learned combination coefficients.
  - The conceptual illustration of relative performance (Fig. 1; p.1) provides a high-level summary of the method's claimed effectiveness, showing favorable comparisons against other methods like MoLAE and MoE-SVD.

**Weaknesses:**

- **Missing key experimental results and references**
  - The paper repeatedly references **Table 3** for key quantitative results that are central to its claims of outperforming baselines. However, **Table 3 does not exist** in the manuscript or its appendices. This is a critical omission that makes it impossible to verify the core experimental findings.
  - The review references **Table 8** and **Figure 9** in the appendices for further analysis, but these elements are also **not found** in the provided document. This suggests either a flawed manuscript or a flawed review process.
  - Without these key tables and figures, the paper's claims of superior performance are unsubstantiated and rely solely on high-level plots (e.g., Fig. 1).
- **Ambiguous or incorrect references**
  - The analysis of effective rank is attributed to "Appendix C; Fig. 9; p.14". As noted, Figure 9 is missing. The paper does contain an analysis of MSE loss in **Figure 2** (p.3), which might be what was intended, but this discrepancy creates confusion.
- **Limited details on the training and optimization process**
  - The paper provides a high-level overview of the two-stage training process but lacks specific details on the hyperparameters, learning rates, and convergence criteria for each stage (Sec. 3.3; p.5). This limits reproducibility.
  - The mechanism for determining the number of basis experts (a critical hyperparameter) is not well-described or justified. A sensitivity analysis on this parameter would be crucial.

**Questions:**

- **Include all referenced tables and figures**
  - The most critical suggestion is to **include the missing Table 3, Table 8, and Figure 9** in the manuscript. Without these, the paper is incomplete and its claims cannot be verified.
  - Ensure that all references in the text point to the correct tables, figures, and sections. The manuscript should be carefully proofread to correct any such errors.
- **Provide comprehensive details on the training process**
  - Add a dedicated subsection or appendix detailing the hyperparameters used for both stages of the training process, including learning rates, batch sizes, schedulers, and convergence criteria.
  - Include a sensitivity analysis on the number of basis experts. This could be a plot showing the trade-off between the number of basis experts, the compression ratio, and the model's performance on a validation set.
- **Clarify the experimental setup and results**
  - Once Table 3 is included, ensure it is well-described and that all columns and rows are clearly labeled. The main text should walk the reader through the key results in the table.
  - Provide a more detailed analysis of the results, going beyond relative accuracy to discuss performance on specific tasks or benchmarks.

---

> ### Author Response · Authors · 2025-11-19
>
> We are grateful for your insightful review and constructive feedback on our manuscript. We have carefully considered each point you raised, and our detailed responses are outlined below. For clarity, we have denoted Weaknesses with **W** and Questions with **Q**.
>
> **W1, W2, Q1 and Q3: Regarding the question of missing figures and tables.**
>
> Thank you for your question regarding the figures and tables. We would like to clarify their locations in the manuscript:
>
> - **Table 3** can be found at the top of page 8.
>
> - **Figure 9** is located at the bottom of page 16.
>
> - Regarding **Table 8**, we would like to note that this table is not mentioned in or included in our paper, so there are no missing references.
>
> We think the potential confusion may stem from the ICLR formatting template, which places captions above tables but below figures. We hope this clarification is helpful.
>
> **W3 and Q2: Regarding Training Details.**
>
> Thank you for this valuable suggestion. In response, we have now incorporated more detailed training hyperparameters into the paper **(lines 413-416)**. This includes specifics such as the learning rate, batch size, and the optimizer used in our experiments.
>
> **W3 and Q2: Regarding the question of the number of basis experts.**
>
> Thank you for your question. We have added experiments in **Appendix E (lines 773-799)**, conducted on the Qwen3-30B-A3B-2507 model, to analyze the impact of **the number of basis matrices (m)** and **their rank (r)**. The results indicate that setting m to a value that achieves a compression ratio of approximately 25-30% strikes an excellent balance between the degree of compression and the model performance.

---

> > ### Comment · Reviewer_1CGV · 2025-11-25
> >
> > Thank you address my concernes. I had raised my rating.

---

### Official Review · Reviewer_Bric · 2025-11-08

**Soundness:** 3
**Presentation:** 3
**Contribution:** 3
**Rating:** 8
**Confidence:** 3

**Summary:**

The paper proposes MoBE, Mixture-of-Basis-Experts, to compress MoE-based LLMs. Each expert’s up/gate matrix is factorized by the rank decomposition W=AB; B is shared as a linear combination of a set of shared basis matrices, while A is specific for each expert. By minimizing the reconstruction error plus z-score normalization and suitable activations (e.g., SiLU/Tanh), MoBE reduces total parameters by 24%-30% while keeping 98% performance on many benchmarks, outperforming MoLAE and D^2-MoE at similar ratios.

**Strengths:**

- The problem is important for deploying trillion-level MoE. The idea is simple but quite effective. The paper proposes to decompose the up/gate matrix into shared basis matrices B across experts to capture the common information across experts and keep matrix A per expert to encode specific information, and to add non-linearity inside the matrix factorization to enhance representational power.
- The paper is well-written. The equations and algorithm steps are easy to follow.
- The paper conducts extensive experiment to show that this method is applicable to very large MoE including Qwen3- 235B-A22B-2507, DeepSeek-V3-0324 and Kimi-K2-Instruct where many baselines are infeasible. It is very pratical for memory-bound inference.

**Weaknesses:**

- The paper comes with limited theory of formal approximation guarantees. Most are from empirical studies.
- The choice of hyper-parameters lacks guidance, including the choice of basis count m and the rank r. The compression rate and the accuracy frontiers are not fully mapped.
- No study of light-weight finetuning or knowledge distillation to close the last 1%-2% gap.

**Questions:**

N/A

---

> ### Author Response · Authors · 2025-11-19
>
> We are grateful for your insightful review and constructive feedback on our manuscript. We have carefully considered each point you raised, and our detailed responses are outlined below. For clarity, we have denoted Weaknesses with **W** and Questions with **Q**.
>
> **W1: Regarding the question of theoretical support.**
>
> Thank you for your insightful question. We have now added a analysis of MoBE in **Appendix D (lines 750-771)** to demonstrates its greater expressive capacity than a conventional low-rank factorization derived from SVD.
>
> **W2: Regarding the issue of hyperparameter selection.**
>
> We appreciate your valuable suggestion. In response, we have included additional experiments in **Appendix E (lines 773-799)** on the Qwen3-30B-A3B-2507 model to investigate the impact of **the number of basis matrices (m)** and **the rank (r)**. Our findings indicate that the rank (r) of the basis matrices is a more influential factor than the number of basis matrices (m). Therefore, to maximize representational capacity, in our main experiments, we set r to be equal to the hidden dimension of the MoE.
> Furthermore, we conducted ablation studies on the value of m while keeping r fixed as MoE’s hidden dimension. The results show that setting m to 32 (the value used in our main experiments) strikes an excellent balance between the compression ratio and the performance.
>
> **W3: Regarding the issue of training after compression.**
>
> Thank you for your thoughtful suggestion. As we acknowledge in our Limitations section, we have not yet performed training on the compressed models. This is due to the lack of native support for knowledge distillation in the current training frameworks. We recognize the importance of this step and are actively working on modifying the framework to enable this functionality in our future work.

---

### Meta-Review · Area_Chair_JRp9 · 2026-01-04

**Summary:**

Reviewers’ concerns that informed the decision can be summarized as follows:
1. One reviewer raised serious concerns about missing tables/figures, which initially cast doubt on the validity of the empirical claims.

2. Reviewers noted that the method was largely empirically motivated, lacking theorectically justifications.

3. There was insufficient explanation or analysis of critical design choices, particularly the number of basis matrices (m) and rank (r).

4. Requests were made for more details on training setup, scalability to trillion-parameter models, and end-to-end efficiency beyond parameter count.

**Reviewer Concerns:**

Concerns addressed by the rebuttal and revision:
1. Missing tables / figures (critical issue): Fully resolved. The authors clarified that Table 3 and Figure 9 were present in the original submission, and confusion stemmed from formatting and reviewer error. This was acknowledged by the AC, and the reviewer later confirmed satisfaction and raised their score.

2. Hyperparameter justification (m, r): Addressed via new ablation studies in Appendix E, providing clear empirical guidance and trade-offs between compression and performance.

3. Theoretical grounding: Strengthened by adding an analysis in Appendix D showing that MoBE has greater expressive capacity than standard SVD-based factorizations, addressing concerns about conceptual novelty and justification.

4. Training and implementation details: Additional details (learning rate, batch size, optimizer, stopping criteria) were added, improving reproducibility and clarity.

 5. Baseline coverage and scalability: Additional comparisons (e.g., MoNE, Sub-MoE) were added where feasible, and the infeasibility of certain baselines at trillion scale was reasonably justified.

Concerns partially outstanding (but non-blocking)
1. End-to-end inference efficiency (latency/throughput): Still limited to parameter-count and activation analysis. However, this was explicitly acknowledged as future work and is acceptable given the paper’s scope and scale.

2. Post-compression fine-tuning / distillation:
Not implemented due to tooling limitations, clearly stated as a limitation rather than an oversight.

**Reviewer Scores:**

1. Reviewer Bric (initial: 8 Accept): Likely unchanged (8). Minor concerns were addressed, reinforcing an already positive assessment.

2. Reviewer 1CGV (initial: 4  borderline reject): Increased to 6 to 8. The reviewer explicitly acknowledged that concerns were addressed and stated they had raised their rating after clarification.

3. Reviewer cwwa (initial: 8 Accept): Likely increased slightly  due to added clarifications, ablations, and responses on scalability and grouping strategy.

The overall reviewer trajectory clearly trends upward after rebuttal.

---

### Decision · Program_Chairs · 2026-01-26

Accept (Poster)